# TASK-ADAPTATION CURRICULUM LEARNING

## ABSTRACT

A large distribution gap between a target task and pre-training tasks could under-mine the task adaptation performance of pretrained models. When the target-task data are scarce, naïve finetuning results in overfitting and forgetting. In various do-mains, skills can be transferred across semantically related tasks, among which the general-purposed ones often have more training data. *Can we bridge the gap be-tween a pre-trained model and a low-resource target task by leveraging data from other tasks?* In this paper, we address the low-resource task adaptation challenge by a transfer learning curriculum, which finetunes a model on a curated sequence of intermediate tasks, thereby progressively bridging the gap between the pre-trained model and the target task. To this end, we formulate the task curriculum as a graph search problem and improve the efficiency of estimating transferability between tasks. Two search algorithms are studied, i.e., greedy best-first search and Monte Carlo tree search. We evaluate our approach, i.e., "task-adaptation cur-riculum learning (TACL)" on two benchmark settings. Extensive evaluations on different target tasks demonstrate the effectiveness and advantages of TACL on highly specific and low-resource downstream tasks.

## 1 INTRODUCTION

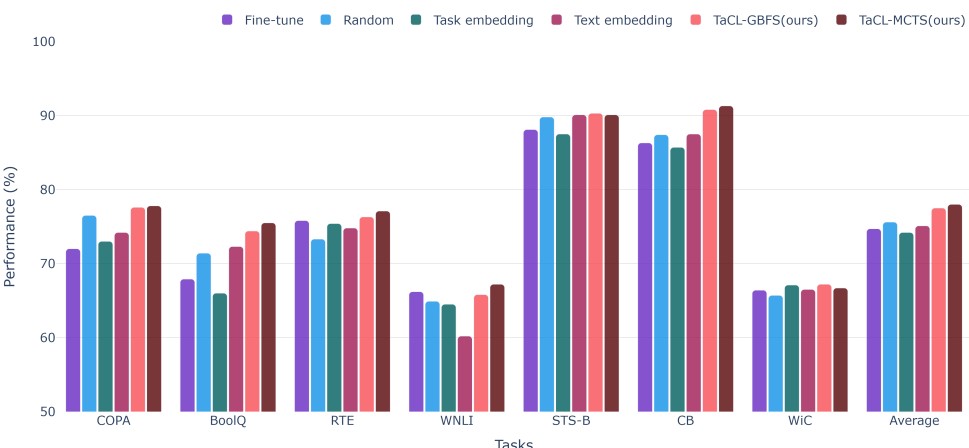

Figure 1: Test accuracy (%) of 7 target tasks (x-axis) achieved by applying six different transfer learning strategies to a graph (or set) of 20 source tasks. TACL (ours) consistently performs the best across all the 7 target tasks, while MCTS outperforms GBFS on $5/7$ target tasks.

Pretrained models have shown a substantial potential to generalize to downstream tasks with promis-ing performance (Peters et al., 2018; Devlin et al., 2019). While finetuning these models on target task data usually suffices for a transfer learning from the pre-trained task(s) to the target task, the final performance heavily depends on the distribution shift between the two tasks and the amount of available data for the target task, since transfer learning may perform poorly under large distribution shift and deficient target task data (Kirkpatrick et al., 2017; Wang et al., 2019).

Fortunately, many downstream tasks are semantically related and their data can be re-formatted for general purposes, so there may exist tasks encapsulating pertinent information for a low-resource

target task. Hence, finetuning a pre-trained model on those intermediate tasks potentially improves the adaptation to the target task (Phang et al., 2019; Vu et al., 2020; Poth et al., 2021) and facilitates smoother knowledge transfer from pre-trained tasks.

Motivated by the efficacy and utility of relevant tasks, we aim to devise a method that guides the model training through a sequence of intermediate tasks. We compare it with conventional transfer learning in Figure 2. There are two potential advantages of this approach: (1) its training data is accumulated along the sequence and thus alleviates the data scarcity of the target task; (2) it establishes a seamless transfer pathway from pretraining tasks to the target task, bridging the distribution gap between them. However, searching for the optimal transfer curriculum presents a formidable challenge, characterized by a combinatorial optimization problem. The impracticality of a brute-force search becomes evident as the sequence length increases, leading to an exponential growth in the number of possible curricula of intermediate tasks. Furthermore, discerning the relative gain or contribution of each task to the target task is non-trivial in practice. Additionally, the dynamic nature of model parameters, altered after training on each task, makes it hard to determine a sequence in advance.

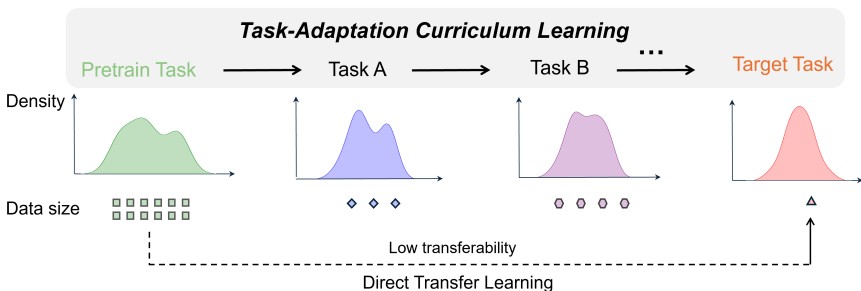

Figure 2: Conventional transfer learning (bottom) vs. task-adaptation curriculum learning (top).

To mitigate these challenges, we formulate the problem as searching for a path on a graph of tasks, effectively connecting the pre-trained task to the target task. This graph-based approach offers several advantages in tackling these issues. Firstly, leveraging existing graph search algorithms allows us to confine the search space, thereby circumventing the need for a computationally intensive brute-force solution. Secondly, the flexibility of employing heuristic or non-heuristic methods facilitates the estimation of the priority of tasks to explore on the graph. Lastly, the dynamic nature of graph search takes into account the evolving model parameters.

To this end, we proposed the framework of *task-adaptation curriculum learning* (TACL), which involves finding a sequence of adaptation tasks that progressively bridges the gap between the pretrained model and the target task by searching a transfer learning path on a graph of tasks. Specifically, we employ two classic search algorithms within this framework: greedy best-first search (GBFS) and Monte-Carlo tree search (MCTS) (Coulom, 2006). Approximation methods are applied to avoid intensive computation. Our approach is examined on two sets of NLP tasks. Through a meticulous analysis of the experimental results, we find that task-adaptation curriculum learning emerges as a beneficial approach, particularly in scenarios with limited data availability. An empirical comparison of different transfer learning strategies on seven target tasks is provided in Figure 1, showcasing the advantages of TACL. Furthermore, our findings underscore the scalability and flexibility of this framework, showcasing its adaptability to diverse task settings.

## 2 RELATED WORK

**Transfer learning and multi-task learning**  The method we propose in this paper addresses the above problem of task adaptation (Zhai et al., 2019; Neyshabur et al., 2020), which generally refers to adapting a pre-trained model to a downstream task. Commonly employed practices include fine-tuning directly and linear probing. Others, such as task/domain-adaptive methods, consider the issue of catastrophic forgetting (Kirkpatrick et al., 2017; Wang et al., 2019), wherein models may forget knowledge from previous tasks after training on a new one, leading to negative transfer. DAPT (Gururangan et al., 2020) tackles this by first tuning the pre-trained model on data related to the

target domain or the target task itself, and then fine-tuning the adaptive-tuned model on the target task. Similarly, Dery et al. (2021) propose a multi-task framework to bridge the gap between pre-trained tasks and the end task by adaptively updating the weights of auxiliary tasks. However, our method differs in that it seeks to design an algorithm capable of automatically determining intermediate training task sequences between pre-trained tasks and the target task, eschewing a multi-task approach.

**Task Transferability**   The concept of intermediate training is also pertinent to our work. In this paradigm, practitioners typically designate one task as an intermediate step between pre-trained tasks and the target task. Previous works in this domain leverage transferability or similarity to identify intermediate tasks (Vu et al., 2020). Estimating task transferability has been a long-studied problem. Past works mainly use Bayesian optimization (Weiss et al., 2016) and information theory (Bao et al., 2019; Tan et al., 2021). LEEP (Nguyen et al., 2020) proposes to apply linear probing to the source-task trained model on the target-task data and uses the performance as a transferability metric. Moreover, task embeddings for transfer learning (Achille et al., 2019) consider the Fisher information matrix of a model fine-tuned on a task as the "task embedding", predicting inter-task transferability by computing the cosine similarity between the task embeddings of the source and target tasks. Notably, our approach diverges in that we seek not just one intermediate task but a sequence of adaptation tasks.

**Curriculum Learning**   Curriculum Learning (CL) was first introduced by Bengio et al. (2009) as a training strategy analogous to the progressive learning nature of humans. A common form of CL is to rank the difficulty or priority of learning examples and then proceed with learning in such a sequence. Subsequent works have further explored this idea by studying different criteria or metrics for data selection. For example, Jiang et al. (2015); Zhou et al. (2020) adjusted the progression pace based on the difficulty of data, and Jiang et al. (2014); Zhou & Bilmes (2018) further take the data diversity into account of curriculum design. Our method also intersects with the concept of curriculum learning. While traditional curriculum learning operates at the data level, our focus in the realm of task adaptation learning is on task-level curriculum learning. Noteworthy work by Pentina et al. (2015) employs curriculum learning to sequentially solve multiple tasks, demonstrating its superiority over joint task-solving. Their aim, however, was to enhance the average performance across multiple tasks, whereas our method specifically targets the performance improvement of the target task.

## 3   PROBLEM FORMULATION

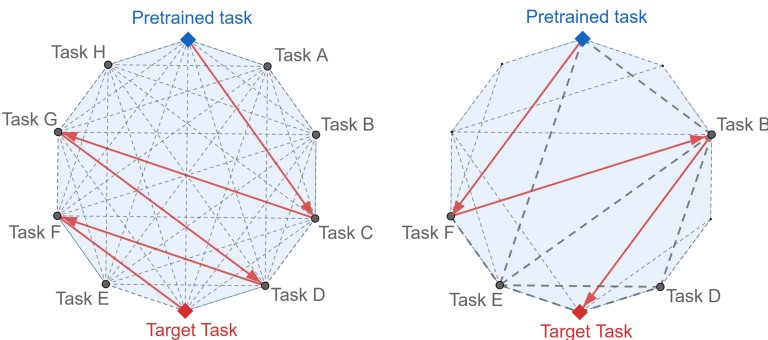

Figure 3: Example of a task-adaptation curriculum on the task graph, which bridges the pre-trained and target tasks by a sequence of intermediate tasks. (**Left**) Searching on a fully connected graph. (**Right**) Searching on a pruned subgraph of the fully connected graph.

Given a target task $\mathcal{T}_t$, our aim is to improve the performance on $\mathcal{T}_t$ by leveraging a set of $n$ candidate source tasks $\{\mathcal{T}_1, \mathcal{T}_2, \mathcal{T}_3, \cdots, \mathcal{T}_n\}$. A task graph, denoted as $\mathcal{G}_n$, is a graph wherein the $n$ nodes represent individual tasks, and the edges symbolize the connections between these tasks. Typically, we assume $\mathcal{G}_n$ to be a complete graph, meaning that each task is directly connected to every other task in the graph. However, $\mathcal{G}_n$ can also be a sparse graph to avoid intensive

computation, illustrated by Figure 3. Now, our objective is to find an optimal sequence $\sigma(\cdot)$ of $l$ intermediate training tasks starting from a pretrained encoder $f(\cdot; \theta)$ with parameters $\theta = \theta_0$, i.e.,

$$s := \text{Pretrained } \theta_0 \to \mathcal{T}_{\sigma(1)} \to \mathcal{T}_{\sigma(2)} \to \cdots \to \mathcal{T}_{\sigma(i)} \cdots \to \mathcal{T}_{\sigma(l)} \to \mathcal{T}_t$$

This sequence is a path on $\mathcal{G}_n$ connecting the pre-trained task to the target task, aiming to maximize the performance of $\mathcal{T}_t$. We continuously finetune the encoder parameter $\theta$ on each intermediate task-$i$ (with learning rate $\eta$), whose loss $\mathcal{L}_{\sigma(i)}(\cdot, \cdot)$ is computed based on the prediction produced by a task-specific output head $\phi_i$. For each task-$\sigma(i)$ in the curriculum, we minimize its loss $\mathcal{L}_{\sigma(i)}$, i.e.,

$$\phi_i, \theta_i \leftarrow \underset{\phi, \theta \in \mathcal{B}(\theta_{i-1})}{\arg\min} \mathcal{L}_{\sigma(i)} \left( \phi[f(x; \theta)], y \right), \tag{1}$$

where $\theta \in \mathcal{B}(\theta_{i-1})$ means that $\theta$ is initialized from the encoder parameters $\theta_{i-1}$ from the last task. We repeat the above procedure from $i = 1$ to $l$ and then finetune the whole model by minimizing the training loss $\mathcal{L}_t^{train}(\cdot, \cdot)$ of the target task-$t$. The optimization of curriculum $s$ can be formulated as a discrete nested optimization problem below, whose top-level objective is the validation-set loss $\mathcal{L}_t^{val}(\cdot, \cdot)$ of the target task.

$$\sigma^* \in \underset{\sigma}{\arg\min} \mathcal{L}_t^{val} \left( \phi_t[f(x; \theta_t)], y \right), \tag{2}$$

$$s.t. \ \ \phi_t, \theta_t = \underset{\phi, \theta \in \mathcal{B}(\theta_l)}{\arg\min} \mathcal{L}_t^{train} \left( \phi[f(x; \theta)], y \right), \tag{3}$$

$$\phi_l, \theta_l \leftarrow \underset{\phi, \theta \in \mathcal{B}(\theta_{l-1})}{\arg\min} \mathcal{L}_{\sigma(l)} \left( \phi[f(x; \theta)], y \right), \tag{4}$$

$$\cdots$$

$$\phi_1, \theta_1 \leftarrow \underset{\phi, \theta \in \mathcal{B}(\theta_0)}{\arg\min} \mathcal{L}_{\sigma(1)} \left( \phi[f(x; \theta)], y \right) \tag{5}$$

To address this optimization problem, we would explore the discrete space consisting of every possible sequence $s$ of tasks defined by $\sigma$. However, a significant number of $\sigma$ are not worth investigating. Therefore, the strategic pruning of unhelpful branches becomes imperative. To achieve this, we adopt the approach of searching on a graph of tasks, dynamically evaluating the value of each task during the search from the current state of the model. This process can be conceptualized as utilizing search algorithms to approximate the outer level of the original optimization problem. In essence, we seek to find the optimal sequence of tasks $s^*$ through a search algorithm, operating on the graph $\mathcal{G}_n$, and simultaneously find the minimum of the target task's training loss $\mathcal{L}_t^{train}(\cdot, \cdot)$. This dynamic and iterative exploration allows us to efficiently prune the solution space of $\sigma$, leading to a more effective and targeted approach to solving the nested optimization challenge.

$$\mathcal{T}_{\hat{\sigma}(i+1)} = \text{SearchAlg}(\mathcal{T}_{\hat{\sigma}(i)}; \mathcal{G}_n)$$

Here, SearchAlg denotes the search algorithm employed to determine the subsequent task in the sequence, given the current task $\mathcal{T}_{\sigma(i)}$. Consequently, the searched sequence $\hat{s}$ is achieved as :

$$\hat{s} := \text{Pretrained } \theta_0 \to \mathcal{T}_{\hat{\sigma}(1)} \to \mathcal{T}_{\hat{\sigma}(2)} \to \cdots \to \mathcal{T}_{\hat{\sigma}(i)} \cdots \to \mathcal{T}_{\hat{\sigma}(l)} \to \mathcal{T}_t$$

## 4 TASK-ADAPTATION CURRICULUM LEARNING (TACL)

In the realm of task-adaptation curriculum learning, our aim is to determine a sequence of adaptation tasks that bridge the gap between the pre-trained task and the target task, with the ultimate goal of enhancing the performance on the target task. Framed as a search problem, we introduce two straightforward yet effective methods: the greedy best first search (GBFS) and Monte-Carlo tree search (MCTS), both geared towards identifying the optimal adaptation sequence.

### 4.1 GREEDY SEARCH OF TASK CURRICULUM

The concept of greedy search, a prevalent technique in the field of search algorithms, involves making the best possible decision at each step. This approach entails examining only the immediate future and selecting the most favorable action. When a problem exhibits an optimal substructure

property, the greedy algorithm tends to yield optimal results. Due to its simplicity and efficiency, greedy algorithms are frequently employed to solve optimization problems.

In task-adaptation curriculum learning, the challenge is to select the subsequent adaptation task after training on a given task. The objective is to make decisions that collectively enhance the overall performance on the target task. In the case of greedy best first search, we adopt a methodical approach by selecting the most promising task at each step. This involves training the model on each auxiliary task, followed by fine-tuning on the target task. Then, the validation accuracy or validation loss on the target task serves as a heuristic value, representing the efficacy of each auxiliary task in aiding the target task. The chosen task is the one that maximizes the estimation of the target task performance. This process is elucidated in detail in algorithm 1 and more discussions on heuristic are in Appendix A.

---

**Algorithm 1** Greedy Best First Search on Task Graph $\mathcal{G}_n$

---

**Require:** $l$: Length of sequence
**Require:** $\mathcal{G}_n$: Task graph with nodes representing tasks
**Require:** $f_\theta$: Pre-trained model
 1: $\mathcal{T}_{\text{current}} \leftarrow$ Source task
 2: **for** $k = 1$ to $l$ **do**
 3:     $\mathcal{N}(\mathcal{T}_{\text{current}}) \leftarrow$ Neighbors of $\mathcal{T}_{\text{current}}$ in $\mathcal{G}_n$
 4:     **for** $\mathcal{T}_i \in \mathcal{N}(\mathcal{T}_{\text{current}})$ **do**
 5:         Train $f_\theta$ on $\mathcal{T}_i$: $\theta_i = \theta - \alpha \nabla_\theta \mathcal{L}(\mathcal{T}_i)$
 6:         Compute heuristic value $h(\mathcal{T}_i)$ on the target task $\mathcal{T}^*$
 7:     **end for**
 8:     $\mathcal{T}' \leftarrow$ Task with the best $h(\mathcal{T}_i)$ from $\mathcal{N}(\mathcal{T}_{\text{current}})$
 9:     Update $\theta \leftarrow \theta - \alpha \nabla_\theta \mathcal{L}(\mathcal{T}')$
10:     $\mathcal{T}_{\text{current}} \leftarrow \mathcal{T}'$
11: **end for**

---

### 4.2 Monte Carlo Tree Search of Task Curriculum

Monte Carlo Tree Search (MCTS) proposed by Coulom (2006) is a heuristic search algorithm designed for decision processes, particularly in applications involving playing board games. In such scenarios, MCTS is employed to solve the intricate game tree by approximating the true game-theoretic value of potential actions from the current state. The algorithm achieves this by iteratively constructing a partial search tree.

A notable advantage of MCTS lies in its independence from domain-specific knowledge, rendering it applicable to a wide range of domains that can be modeled using a tree structure. In the realm of task-adaptation curriculum learning, the process of determining the next task inherently involves decision-making, akin to a growing tree structure. Consequently, MCTS seamlessly aligns with our framework for task-adaptation curriculum learning, offering a versatile and domain-agnostic approach to solving the intricate decision processes involved in the selection of intermediate tasks. In this context, the state represents the current model, a node corresponds to a specific task, an action involves training on the chosen task, and the reward is determined by the performance of the target task after completing the adaptation sequence. A simulation entails training the model on a sequence of tasks of a specified length.

How the tree is built depends on how nodes in the tree are selected. By framing the choice of a child node as a multiarmed-bandit problem, we employ the Upper Confidence Bound (UCB1) algorithm to estimate the value of each child node. The UCB1 algorithm considers the expected reward as approximated by Monte Carlo simulations, treating these rewards as random variables with unknown distributions. This approach ensures simplicity, efficiency, and a guaranteed closeness to the best possible bound on the growth of regret. The selection of a child node is determined by the following formula:

$$v' := \underset{v' \in \text{ children of } v}{\arg\max} \frac{Q(v')}{N(v')} + c\sqrt{\frac{2 \log N(v)}{N(v')}}. \tag{6}$$

| Task | \| Train \| | Task type | Domain |
|---|---|---|---|
| MNLI (Williams et al., 2018) | 393K | NLI | misc. |
| QQP (Iyer et al., 2017) | 364K | paraphrase | social QA |
| QNLI (Wang et al., 2018) | 105K | QA-NLI | Wikipedia |
| SNLI (Bowman et al., 2015) | 570K | NLI | misc. |
| SST-2 (Socher et al., 2013) | 67K | sentiment analysis | movie reviews |
| CoLA (Warstadt et al., 2019) | 8.5K | grammatical acceptability | misc. |
| STS-B (Cer et al., 2017) | 7K | semantic similarity | misc. |
| MRPC (Dolan & Brockett, 2005) | 3.7K | paraphrase identification | news |
| RTE (Dagan et al., 2005) | 2.5K | NLI | news, Wikipedia |
| WNLI (Levesque et al., 2012) | 634 | coreference NLI | fiction books |
| SQuAD (Rajpurkar et al., 2016) | 108K | QA | Wikipedia, crowd |
| DROP (Dua et al., 2019) | 77K | reading comp. | Wikipedia, crowd |
| WikiHop (Welbl et al., 2018) | 51K | multi-hop QA | Wikipedia, KB |
| BoolQ (Clark et al., 2019) | 16K | natural yes/no QA | Wikipedia, web queries |
| CQ (Bao et al., 2016) | 2K | knowledge-based QA | snippets, web queries/KB |
| WiC (Pilehvar & Camacho-Collados, 2019) | 5.4K | word sense disambiguation | misc. |
| COPA (Roemmele et al., 2011) | 400 | commonsense reasoning | blogs, encyclopedia |
| CB (De Marneffe et al., 2019) | 250 | NLI | various |
| WSC (Levesque et al., 2012) | 554 | coreference resolution | fiction books |
| ANLI (Nie et al., 2020) | 163K | NLI | misc. |

Table 1: Summary of the tasks and their datasets used in our experiments.

Here, $N(v)$ is the number of times the current (parent) node has been visited, $N(v')$ is the number of times the child has been visited, and $c > 0$ is a constant.

As a result, we employ UCB1 for the selection process and implement a random policy for rollout. The performance of the target task, such as validation accuracy or loss, is utilized to compute the reward associated with a given sequence. As the tree grows, we iteratively refine our estimates of the value of choosing the next task. The entire process is encapsulated in algorithm 2 in appendix.

## 5 EXPERIMENTS

In our experimental investigations, we aim to address the following questions pivotal to the efficacy of our proposed task-adaptation curriculum learning (TACL) methodology: (1) Can models gain significant benefits from the adoption of TACL? (2) What are some similarities and differences in the results produced by GBFS and MCTS? (3) What are some possible factors that could potentially influence the performance of TACL?

### 5.1 EXPERIMENTAL SETTING

To systematically address these questions, we designed and conducted experiments on two graphs: a smaller graph comprising 6 tasks and a larger graph encompassing all 20 tasks. This experimental setup enables us to evaluate the robustness and scalability of our proposed approach under varying parameter settings. We selected 20 representative NLP tasks spanning diverse categories and requiring different types of knowledge, as detailed in Table 1. These categories include natural language inference, question answering, reading comprehension, sentiment analysis, etc. The diverse nature of these datasets allows us to comprehensively evaluate the adaptability of our method across various NLP tasks.

### 5.2 BASELINES

**Fine-tune:** One of our baseline comparisons involves the direct fine-tuning of the model, as this serves as a standard approach and aligns with our primary goal of enhancing the performance of fine-tuning on the target task.

**Random:** In addition to direct fine-tuning, we include a random sequence of the same length as the paths searched by our method as an additional baseline. This comparison aims to evaluate whether

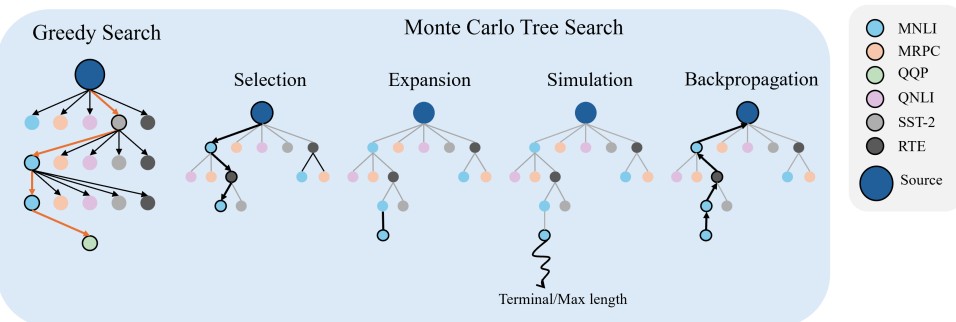

Figure 4: Greedy search vs. Monte Carlo tree search on searching a curriculum for target task QQP.

our method can effectively discover valuable information regarding task transferability within the graph, as opposed to a random exploration.

**Task/text embedding:** We also explore two common methods for estimating transferability between tasks, which can aid in finding an intermediate task. These methods involve mapping tasks to embeddings/vectors (Achille et al., 2019; Vu et al., 2020) and utilizing cosine similarity between these embeddings to estimate transferability.

### 5.3 TACL ON A 6-TASK GRAPH

In this experimental setup, we use six tasks from the GLUE benchmark (Wang et al., 2018) and the BERT model (Devlin et al., 2019). Additionally, every task included in this graph is considered a potential target task, allowing for comprehensive exploration and evaluation of the model's adaptability across various tasks. The core aim of our experiments is to evaluate the efficacy of TACL in addressing challenges associated with fine-tuning, particularly in situations marked by limited training data. To achieve this, we explore varying levels of data scarcity across different tasks. This diverse range of data limitations enables us to systematically assess the adaptability and performance of our proposed methodology across varying degrees of data scarcity. Further experimental details are in the appendix C.

| Tasks
\| **Train** \| | SST-2
128 | MRPC
128 | MNLI
1K | QNLI
1K | QQP
1K | RTE
2K | Average |
|---|---|---|---|---|---|---|---|
| Fine-tune | 81.8 | 81.2 | 60.2 | 78.6 | 70.6 | 68.6 | 73.5 |
| Random | 74.0 | 80.1 | 62.1 | 77.8 | 71.7 | 70.1 | 72.6 |
| Task embedding | 73.5 | 72.1 | 61.8 | 76.7 | 69.1 | 68.2 | 70.2 |
| Text embedding | 78.3 | 81.0 | 59.4 | 78.8 | 69.2 | 71.1 | 73.0 |
| TACL-GBFS (ours) | 84.2 | **83.2** | **64.9** | 79.0 | 73.0 | 71.5 | 76.0 |
| TACL-MCTS (ours) | **85.0** | 83.1 | 64.2 | **79.9** | **73.6** | **72.8** | **76.4** |

Table 2: Target task's test-set performance (%) achieved by different transfer learning strategies on a small graph of six-tasks.

Table 2 presents the results for each task treated as the target task. These results reflect the performance of a fully converged model on the target task. The limitations imposed by the scarcity of data make direct fine-tuning ineffective, resulting in suboptimal outcomes. Random sequences sometimes exhibit slightly improved results, aligning with the understanding that incorporating intermediate training tasks in data-limited scenarios can offer some benefits. For most target tasks, embedding methods struggle to capture the relative importance of auxiliary tasks, leading to unsatisfactory results. In contrast, our proposed methods demonstrate significant success in enhancing the performance of the target task across all tasks in the graph. Notably, Monte Carlo Tree Search (MCTS) outperforms Greedy Best-First Search (GBFS) in most tasks, indicating that the iterative

nature of MCTS likely contributes to its superior performance in navigating the task graph and identifying more effective adaptation sequences. This observation underscores the effectiveness of our task-adaptation curriculum learning framework in comparison to baseline methods.

**Analysis of paths and structures within the task graph**    In addition to evaluating performance, our investigation aims to determine whether our method can uncover specific structures within the graph that are relevant to the target task. Figure 5 depicts the paths discovered by Greedy Best-First Search (GBFS) to all target tasks. Figure 6 demonstrates some paths to QQP by Monte Carlo tree search. While the paths are not entirely deterministic due to the choice of random seed, we are still able to discover some important patterns and structures within the graph of tasks.

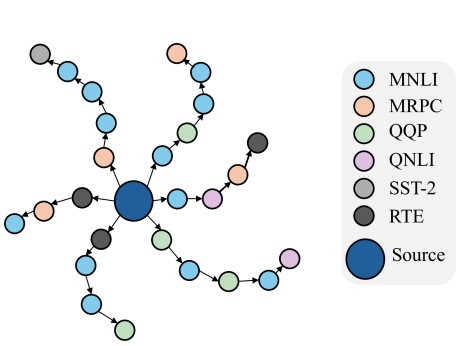

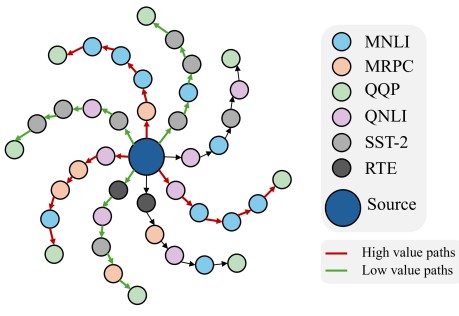

Figure 6: Comparing candidate task-curricula for QQP (target task) in Monte Carlo tree search. Better curricula with higher values are highlighted in red.

Figure 5: TACL-curricula for six target tasks achieved by greedy best-first search.

As shown in Figure 5, MNLI is the most frequently chosen task in task sequences, and its placement at the end of the sequence may be crucial for the performance on the target task. Furthermore, MNLI tends to be associated with high-value paths produced by MCTS, as illustrated in Figure 6. Multi-Genre Natural Language Inference (MNLI) is a large-scale entailment classification task. In MNLI, given a pair of sentences, the objective is to predict whether the second sentence entails, contradicts, or is neutral with respect to the first one. Based on these observations, we can formulate a hypothesis that the model becomes more proficient in processing and analyzing semantic information after training on MNLI. The frequent inclusion of MNLI suggests its importance in enhancing the model's ability to understand and reason about semantic relationships between sentences. This enhanced capability is expected to translate into improved performance on target tasks. A more detailed analysis can be found in appendix C.

## 5.4   TACL ON A 20-TASK GRAPH

After validating the effectiveness of TACL on a relatively small graph, our aim is to extend our method to a larger graph to assess its flexibility and scalability. When applying it to a graph with numerous tasks, a key concern is minimizing computational costs, particularly given the inherent expense of searching on a fully connected graph, where the number of edges grows quadratically with the number of tasks.

Fortunately, prior research (Vu et al., 2020; Poth et al., 2021; Kim et al., 2023) has extensively explored similarities and transferability among NLP tasks. Leveraging this knowledge allows us to perform clustering and prune edges that may lead to negative transfer. This approach enables us to conduct searches on pruned subgraphs of the original fully connected graph, substantially reducing computational overhead. When such information is not provided, we can efficiently estimate transferability using existing training-free or light-training based methods.

In our experiment, we first constructed a pairwise transferability matrix for all 20 tasks based on previous studies (Vu et al., 2020; Poth et al., 2021). Next, we sparsified the graph by removing edges below a set threshold in the transferability scores. Finally, we constructed subgraphs for target tasks based on these scores. We used the DeBERTaV3 (He et al., 2021) model throughout the experiment

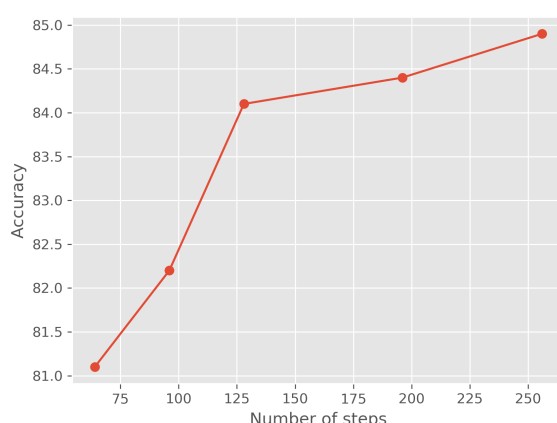

Figure 7: Test accuracy (%) on target-task SST-2 when spending different numbers of training steps on each task in the curriculum.

and limited the training samples to simulate a low-data scenario. More experimental details can be found in Appendix C. As shown in Figure 1, our results indicate that MCTS consistently outperforms all other methods, and greedy search also tends to yield better results. This demonstrates the effectiveness and scalability of TACL.

## 6 DISCUSSION

**Computational cost.** While our method offers considerable efficiency gains compared to an exponential-time brute force approach, TACL encounters a computational bottleneck when estimating the importance of next states during search. This challenge arises due to the high-dimensional nature of pretrained models, which significantly escalates training costs. To adress these problems, when training on an intermediate task within the sequence, we limit the training steps rather than allowing the model to fully converge. This strategy is employed to strike a balance between searching efficiency and obtaining meaningful insights from the intermediate tasks. In the context of Monte Carlo Tree Search (MCTS), simulations can be computationally intensive as they involve iterative fine-tuning of the model. To mitigate this, we reduce the number of steps during simulation, aiming for a more efficient approximation of the true performance.

**Influences of training steps in TACL.** In addition to the final results of TACL, our curiosity extends to understanding the factors that may affect the performance of TACL. Throughout the course of experiments, we observe that the number of training steps on each task within the task sequence is sometimes important in determining the final results. For a fixed sequence of tasks, varying the number of training steps can lead to different outcomes. As depicted in Figure 7, more training steps may help the model in acquiring and preserving more knowledge from the task, resulting in greater improvements on the target task. This observation emphasizes the importance of this hyperparameter to the effectiveness of TACL.

## 7 CONCLUSION

In summary, we have introduced the framework of task-adaptation curriculum learning as a solution to challenges associated with directly fine-tuning pre-trained models. Our approach offers several advantages: it is both simple and flexible, allowing for the incorporation of various search algorithms on graphs. Furthermore, it serves as an extension of intermediate training, leveraging a broader set of tasks to enhance the model's generalizability, particularly in scenarios with limited data.

The adaptability provided by a sequence of tasks may play a crucial role in addressing the disparity between a pre-trained model and a highly specific downstream task. We believe that our methodology contributes some insights to the realm of task adaptation in NLP.

## 8 LIMITATIONS

While our method is evaluated across multiple domains in this study, the diversity of task types examined remains limited. Moreover, our experiments are conducted using relatively small models compared to contemporary large language models (LLMs). Thus, there is an opportunity for future research to extend our method and experiments to encompass a broader range of task types and incorporate larger models. Additionally, further exploration into the influences of hyperparameters within the method could enhance our understanding of its performance.

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

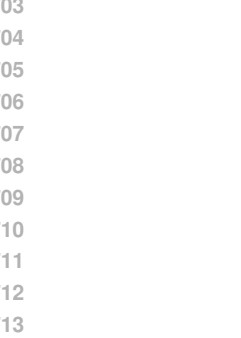

Figure 8: Performance scores (%) of three target tasks achieved by GBFS/MCTS-searched curriculum learning on a nine-task graph. Scores refer to accuracy or F1 score. MCTS-curriculum achieves the best performance, while both MCTS and GBFS outperform direct finetuning.

## A   ALGORITHM DETAILS

The goal of a heuristic is to generate a solution within a reasonable time frame that is sufficiently effective for solving the given problem. The way we compute the heuristic in our algorithms is crucial to addressing the problem. In our approach, determining the heuristic value typically involves first training the model on each auxiliary task, followed by fine-tuning on the target task. Metrics such as loss, accuracy, or F1 score on the target task can then be used as the heuristic value. However, this process can be computationally intensive, as pretrained models often contain millions to billions of parameters. To mitigate this issue, we can reduce the number of training steps or run the process in parallel. Additionally, first-order approximations can be employed to estimate the heuristic value. Alternative metrics, such as those related to model complexity, may also serve as heuristics; however, we leave this avenue for future exploration.

## B   MORE RESULTS ON GLUE BENCHMARK

In the extension of the experiment on the six-task graph, we expanded the graph to include three additional GLUE tasks (STS-B, CoLA, WNLI), resulting in a total of nine tasks. In this case, we focused on observing the performance of our method on three specific tasks: MRPC, QNLI, and RTE. The experimental settings remained consistent with the smaller graph experiment, ensuring a fair comparison.

The results of the experiments are presented in figure 8. As indicated by the results, TACL appears to derive some benefits from a more diverse range of available auxiliary tasks, with slightly improved performance. Upon examining the new paths, it is noteworthy that STS-B is often included in the sequence of adaptation tasks. The Semantic Textual Similarity Benchmark involves sentence pairs sourced from news headlines and other texts, annotated with a score indicating the semantic similarity between the two sentences on a scale from 1 to 5. Given the nature of the STS-B task, which assesses the general semantic knowledge of a model, we can hypothesize that the universal knowledge acquired during the learning process of STS-B may contribute to the model's improved adaptability and performance.

## C   EXPERIMENTAL DETAILS

**More experimental details on the 6-task graph**   Given that the test sets of GLUE datasets are not publicly available, our reported performance metrics are based on the validation sets. We split some samples from the training set to serve as a validation set during the course of our experiments. Regarding performance metrics, we report F1 scores for QQP and MRPC, and accuracy scores for the other tasks. Regarding task embedding and text embedding baselines, our experimental settings closely align with those outlined by Vu et al. (2020). In terms of the training methodology, we

---

**Algorithm 2** Monte Carlo Tree Search

---

**Require:** $\{\mathcal{T}_n\}$: A set of $n$ auxiliary tasks
**Require:** $f_\theta$: Current model
 1: **function** MCTS($f_\theta$)
 2:  **while** within computation budget **do**
 3:   $\mathcal{T}_l \leftarrow$ TREEPOLICY($f_\theta$, null)
 4:   $r \leftarrow$ SIMULATE($\mathcal{T}$)
 5:   BACKUP($\mathcal{T}_l, r$)
 6:  **end while**
 7:  **return** $\arg\max_\mathcal{T}$ UCT(null, 0)
 8: **end function**
 9: **function** TREEPOLICY($f_\theta, \mathcal{T}$)
10:  **while** $\mathcal{T}$ is nonterminal **do**
11:   **if** $\mathcal{T}$ not fully expanded **then**
12:    Choose an untried tasks $\mathcal{T}'$
13:    Add a new child $\mathcal{T}'$ to $\mathcal{T}$
14:    Train $f_\theta$ on $\mathcal{T}'$: $\theta \leftarrow \theta - \alpha\nabla_\theta\mathcal{L}(\mathcal{T}')$
15:    **return** $\mathcal{T}'$
16:   **else**
17:    $\mathcal{T} \leftarrow \arg\max_\mathcal{T}$ UCT($\mathcal{T}, c$)
18:    Train $f_\theta$ on $\mathcal{T}$: $\theta \leftarrow \theta - \alpha\nabla_\theta\mathcal{L}(\mathcal{T})$
19:   **end if**
20:  **end while**
21:  **return** $\mathcal{T}$
22: **end function**
23: **function** SIMULATE($\mathcal{T}$)
24:  **while** $\mathcal{T}$ is nonterminal **do**
25:   Choose $\mathcal{T}'$ randomly
26:   Train $f_\theta$ on $T$: $\theta \leftarrow \theta - \alpha\nabla_\theta\mathcal{L}(\mathcal{T})$
27:   $\mathcal{T} \leftarrow \mathcal{T}'$
28:  **end while**
29:  Train $f_\theta$ on $\mathcal{T}^*$: $\theta \leftarrow \theta - \alpha\nabla_\theta\mathcal{L}(\mathcal{T})$
30:  $r \leftarrow$ evaluate $f_\theta$ on $\mathcal{T}^*$
31:  **return** $r$
32: **end function**
33: **function** BACKUP($\mathcal{T}, r$)
34:  **while** $\mathcal{T} \neq$ null **do**
35:   $N(\mathcal{T}) \leftarrow N(\mathcal{T}) + 1$
36:   $Q(\mathcal{T}) \leftarrow Q(\mathcal{T}) + r$
37:   $\mathcal{T} \leftarrow$ parent of $\mathcal{T}$
38:  **end while**
39: **end function**
40: **function** UCT($\mathcal{T}, r$)
41:  **return** $\frac{Q(v')}{N(v')} + c\sqrt{\frac{2\log N(v)}{N(v')}}$
42: **end function**

---

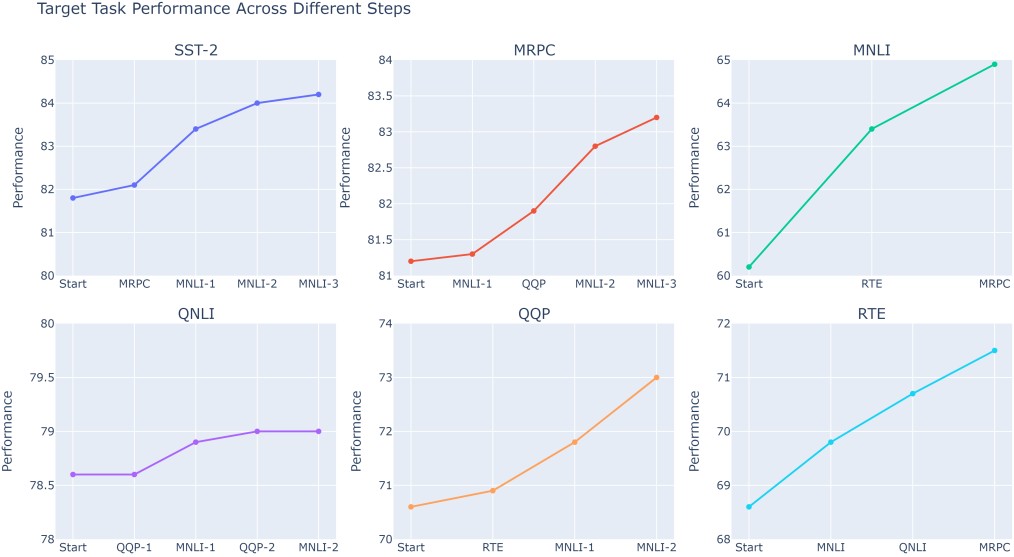

Figure 9: Performance scores (%) across different steps of all target tasks achieved by greedy curriculum on the 6-task graph. Scores refer to accuracy or F1 score.

| Parameter | 6-task graph | 20-task graph |
|---|---|---|
| Checkpoint | `bert-base-uncased` | `microsoft/deberta-v3-base` |
| Max sequence length | 4 | 5 |
| Max steps | 96, 128, 256 | 128, 256 |
| Learning rate | $2 \times 10^{-5}$ | $2 \times 10^{-5}$ |
| Batch size | 16 | 8, 16 |
| Weight decay | 0.01 | 0.01 |
| Learning rate decay | Linear | Linear |
| Adam $\epsilon$ | $1 \times 10^{-6}$ | $1 \times 10^{-6}$ |
| Adam $\beta_1$ | 0.9 | 0.9 |
| Adam $\beta_2$ | 0.999 | 0.999 |

Table 3: Hyper-parameters for experiments on the 6-task and 20-task graphs

use a fresh optimizer for each phase of training. For each task, we add only a single task-specific, randomly initialized output layer to the pre-trained Transformer model. For all experiments, the loss function is the cross-entropy error between the predicted and true class. The implementation is carried out using Hugging Face's transformers library (Wolf et al., 2019) and PyTorch (Paszke et al., 2019). While we follow the recommended hyperparameters by Devlin et al. (2019), we adjust the batch size to suit our experimental requirements.

We also provide figure 9 to illustrate how target task performance evolves across different stages of the curriculum. As the chart indicates, for most tasks (SST-2, MRPC, QNLI, QQP), MNLI contributes the most to performance improvement. For the remaining tasks (MNLI, RTE), MRPC also plays a significant role. MNLI and MRPC are both natural language understanding tasks that focus on semantic relationships between sentence pairs, making them highly relevant for transfer to many target tasks in NLP. To be more specific, MNLI requires the model to understand fine-grained semantic relationships such as entailment, contradiction, and neutrality, providing generalized language understanding and reasoning capabilities that benefit a wide range of target tasks. MRPC, on the other hand, focuses specifically on identifying whether two sentences are paraphrases. This task improves the model's ability to detect semantic equivalence, which is particularly useful for tasks like textual entailment (e.g., RTE).

**More experimental details on the 20-task graph** To reduce computational cost, we conducted the search on subgraphs. Subgraphs were constructed for each task based on its five nearest neighbors (i.e., the top five source tasks as determined by transferability scores). All other conditions remained consistent with the experiments on the six-task graph. For STS-B, we report the Spearman correlation, and for all other tasks, we report accuracy.

| Tasks | COPA | BoolQ | RTE | WNLI | STS-B | CB | WiC |
|---|---|---|---|---|---|---|---|
| \| **Train** \| | 300 | 1K | 1K | 500 | 1K | 200 | 1K |

Table 4: Number of training samples for target tasks in the 20-task graph experiments

