# OpenReview forum: "Task-Adaptation Curriculum Learning"
_ICLR.cc/2025/Conference — ICLR 2025 Conference Withdrawn Submission_

### Official Review · Reviewer_DFHu · 2024-11-01

**Soundness:** 2
**Presentation:** 3
**Contribution:** 2
**Rating:** 5
**Confidence:** 4

**Summary:**

The paper discusses the challenge of adapting pre-trained models to low-resource target tasks, especially when there is a significant distribution gap between the pre-training tasks and the target task. To address this, the authors propose a transfer learning curriculum approach called "task-adaptation curriculum learning (TaCL)" that fine-tunes the model on a sequence of intermediate tasks, progressively bridging the gap between the pre-trained model and the target task. The task curriculum is formulated as a graph search problem, and the paper studies two search algorithms: greedy best-first search and Monte Carlo tree search. The effectiveness of TaCL is evaluated on benchmark settings, showing its advantages in adapting to highly specific and low-resource downstream tasks by leveraging data from other semantically related tasks.

**Strengths:**

1. This paper is well-written and has a good motivation.
2. This paper investigates the challenge of adapting pre-trained models to low-resource target tasks, which is an important and interesting problem that may greatly benefit the deep learning community.
3. This paper formulated the task curriculum as a graph search problem,  which gives a fresh perspective for transfer learning.

**Weaknesses:**

1. The paper uses two search algorithms: greedy best-first search and Monte Carlo tree search. Both of these algorithms are proposed by the existing works, limiting the proposed method's novelty.
2. The proposed task-adaptation curriculum learning (TaCL) is quite similar to the existing work "Don't Stop Pretraining: Adapt Language Models to Domains and Tasks", a more thorough analysis and comparison with it will be favored, especially in the experiment section.
3. This paper proposed a sequential strategy to fully exploit the existing tasks. What about a parallel strategy? For example, if we have six auxiliary tasks, we can fine-tune the first two tasks and then the next four tasks, rather than fine-tune them one by one. Will such a parallel strategy perform better? Further, we can also finetune the six auxiliary tasks together and then on the target tasks. Will such a strategy alleviate forgetting better?
4. The proposed task-adaptation curriculum learning (TaCL) is much heavier than the existing transfer learning methods since it has to train on several extra tasks.  How much extra training time or cost will it bring? What about the return on investment?

**Questions:**

Please refer the weaknesses section.

---

> ### Author Response · Authors · 2024-11-26
>
> Thank you for your review! Our responses to your questions are as follows:
>
> > **Q1**: Can authors provide comparison with DAPT/TAPT.
>
> DAPT/TAPT continues pretraining directly on the target domain/task, which is not feasible in our low-resource setting. It is only possible to continue pretraining on relavant tasks and our baselines do reflect this scenario.
>
> > **Q2**: Why not finetuning more than one task at the same time instead of finetuning them one by one?
>
> Fine-tuning multiple tasks simultaneously could introduce challenges in balancing the contributions of each task and potential negative transfer. Our sequential approach ensures that each task is fine-tuned in a way that maximally benefits the target task, allowing for a more focused and efficient transfer of knowledge.
>
>
> > **Q3**: The proposed method is computationally intensive.
>
> Our method involves additional computational steps, primarily due to the search process. However, we have used optimization with as task graph pruning and limiting search steps, to reduce computational overhead. Additionally, the significant performance improvements achieved on the target tasks demonstrate that the computational investment is worthwhile. Moreover, both greedy search and MCTS are parallelizable in our setting, which can substantially reduce the overall training time. We will explore such strategies to further improve computational efficiency in future work.

---

> > ### Comment · Reviewer_DFHu · 2024-12-03
> > **Response to the authors**
> >
> > Thanks for your feedback.  After reading reviews from other reviewers and the authors' responses, I believe the current submission requires more effort. Therefore, I will maintain my score. Good Luck!

---

### Official Review · Reviewer_imo4 · 2024-11-03

**Soundness:** 3
**Presentation:** 3
**Contribution:** 3
**Rating:** 5
**Confidence:** 3

**Summary:**

This paper explores low-resource task adaptation using multiple auxiliary tasks within a transfer learning curriculum. In this framework, a sequence of auxiliary tasks is selected for model fine-tuning. The authors formulate the task selection process as a graph search problem, and propose two search algorithms to estimate transferability and select tasks. Experiments demonstrate the effectiveness of these algorithms in multi-task transfer learning scenarios.

**Strengths:**

* The paper explores the question of how to effectively select a sequence of tasks for low-resource task adaptation, a novel approach in the few-shot learning domain.
* Experiments demonstrate that the task sequence selection methods outperform full fine-tuning, providing valuable insights into the transfer learning field.
* The paper is clear structured and well-written.
* The authors emphasize the issue of computational cost and propose several improvements to address it.

**Weaknesses:**

The main weakness of the paper is the significance and computational burden of search algorithm.

* This paper explores of how to leverage data from auxiliary tasks for task adaptation. To address the problem, the authors consider a transfer learning curriculum framework and propose some algorithm to select task sequence. However, in each step of the sequential process, a model can learn from only one task. A more simple and straightforward approach is joint learning multi-tasks with adaptive weights for each tasks. In the joint learning process, multiple tasks can interact with each other to improve model performance.
* Another problem is the computational burden of proposed algorithms. Although authors emphasize the issue of computational cost and give some qualitative analysis in discussion, quantitative analysis about full-finetuning and proposed search algorithms is more important. As different strategies involves different training process (e.g. training steps and number of training samples).

**Questions:**

See my comments under weaknesses section.

Another question is as follows:
* Can the authors provide more details about the task embedding methods used in the experiments? Do these methods select only one intermediate task, or do they choose a sequence of tasks, similar to GBFS and MCTS? Additionally, why do the task embedding methods underperform GBFS and full fine-tuning?

---

> ### Author Response · Authors · 2024-11-26
>
> Thank you for your suggestions! Please see our responses below.
>
> > **Q1**: Why not joint learning multi-tasks with adaptive weights for each task?
>
> Joint learning with adaptive task weights is a reasonable multi-task approach. However, we focus on transfer learning so our goal is to improve the performance of target-task only, while the negative transfer in multi-task learning may hinder the target task performance. Without complicated trade-off to balance all tasks, our curriculum in each stage selects tasks that can lead to the maximal gain on the target task.
>
> Joint learning also require extensive exploration and an extra validation set to find the optimal weights. With increasing tasks, their required cost drastically grows so they cannot adpat to low-resource setting. In contrast, our method only needs to search for a task per step to improve the validation set performance for a target task.
>
> > **Q2**: Can authors provide quantitative analysis about the computation burden of the proposed method?
>
> **A2**: The total number of training steps for the curriculum search is approximately 2 to 3.5 times the fine-tuning steps. However, both greedy search and MCTS are parallelizable, which can substantially reduce the overall training time. We will continue exploring such strategies to further improve computational efficiency in future work.
>
> > **Q3**: Can the authors provide more details about the task embedding method?
>
> **A3**: For task embedding, we followed the implementation of Vu et al., selecting one intermediate task. Its suboptimal performance may stem from the approximation methods used, such as considering only the diagonal entries of the Fisher information matrix and relying on empirical estimates of Fisher information. These approximations might result in the loss of critical information.

---

### Official Review · Reviewer_RpUv · 2024-11-03

**Soundness:** 2
**Presentation:** 3
**Contribution:** 2
**Rating:** 3
**Confidence:** 3

**Summary:**

The paper addresses the problem of model fine-tuning, aiming to bridge the gap between a pre-trained model and the low-resource target task. The authors propose to leverage other semantic relevant tasks to improve the target task performance. A task-adaptation curriculum learning (TACL) is proposed to construct a sequence of tasks to enhance fine-tuning. The task sequence selection is formulated as a graph search problem, whereas the greedy search and Monte Carlo tree search are investigated and evaluated. Experiments on two benchmarks are conducted to validate its effectiveness.

**Strengths:**

1. The idea of building task curriculum to enhance fine-tuning performance on target task is interesting.
2. The paper is generally well-written and easy to follow.
3. The target performance of TACL is better than baselines.

**Weaknesses:**

1. While learning from task curriculum may improve the performance on the final target task, it may raise concerns about more severe forgetting and safety risks. For example, previous research [1] shows that fine-tuning may compromise the model safety. The reviewer concerns that the proposed method may exacerbate the problem by introducing a longer fine-tuning path. Therefore, it is suggested that the author should add discussion and experiments on these aspect to validate the method more comprehensively.
2. The analysis of the search results of the six-task graph (fig 5 and fig 6) is insufficient. In the analysis provided, the authors only highlight the importance of a particular task MNLI, whereas the effect of other tasks are left without discussion. Due to the reason, it is still unclear what is the key aspect of the task curriculum that leads to performance boost.
3. In the related work section, the relevant works cited are generally published years ago. It is suggested that the authors include more recent papers.
Ref: [1] Fine-tuning aligned model compromises safety, even when users do not intend to!

**Questions:**

Please refer to the weakness part.

---

> ### Author Response · Authors · 2024-11-26
>
> Thank you for your suggestions! Please see our responses below.
> > **Q1**: The proposed method may exacerbate forgetting and safety risks by introducing a longer fine-tuning path.
>
> **A1**: Forgetting and safety risks are valid concerns when evaluating a general-purpose LLM. However, we focus on transfer learning which aims at training a model for the target tasks only. We will add more relevant discussions.
>
> > **Q2**: The analysis of the search results of the six-task graph is insufficient and it is unclear what is the key aspect of the task curriculum that leads to performance boost.
>
> **A2**: The search algorithms used in TaCL naturally produce interpretation to the values of each task in the curriculum. Since the final improvement is a result of training on a sequence of tasks, model performance in each step depends on all the previous tasks, and each task's contribution cannot be entirely disentangled from others. For simplicity, we highlight several different paths and their values achieved in the search. We will add more details in the next version.
>
> To analyze the key aspect of the curriculum, we added Figure 9 in the appendix to illustrate how target task performance evolves across different stages of the curriculum. As the chart indicates, for most tasks (SST-2, MRPC, QNLI, QQP), MNLI contributes the most to performance improvement. For the remaining tasks (MNLI, RTE), MRPC also plays a significant role. MNLI and MRPC are both natural language understanding tasks that focus on semantic relationships between sentence pairs, making them highly relevant for transfer to many target tasks in NLP.
>
> To be more specific, MNLI requires the model to understand fine-grained semantic relationships such as entailment, contradiction, and neutrality, providing generalized language understanding and reasoning capabilities that benefit a wide range of target tasks. MRPC, on the other hand, focuses specifically on identifying whether two sentences are paraphrases. This task improves the model's ability to detect semantic equivalence, which is particularly useful for tasks like textual entailment (e.g., RTE).
>
> > **Q3**: The relevant works cited are generally published years ago.
>
> **A3**: Thank you for your suggestions. We will add discussions about more recent works. [1] uses transferability score to select source tasks, and this work is cited in our paper. [2] leverages in-context learning to predict transferability between tasks, which is also relevant to our work.
>
> [1] Taskweb: Selecting better source tasks for multi-task nlp. 2023
>
> [2] BenTo: Benchmark Task Reduction with In-Context Transferability. 2024

---

### Official Review · Reviewer_JTSV · 2024-11-03

**Soundness:** 2
**Presentation:** 2
**Contribution:** 2
**Rating:** 3
**Confidence:** 3

**Summary:**

This paper formulates the task curriculum as a graph search problem, aiming to identify a sequence of intermediate tasks that bridge the gap between a pre-trained model and a low-resource target task. Methodologically, the approach integrates two classic search algorithms into its framework: greedy best-first search (GBFS) and Monte Carlo tree search (MCTS). Experimental results on two NLP task sets demonstrate the proposed method's superiority over other relevant baselines.

**Strengths:**

1. The writing is clear and easy to follow.
2. The proposed method, TaCL, which leverages graph search as a curriculum for task adaptation, appears to be valid.

**Weaknesses:**

1. The contributions of this work are vague, as the idea of treating the task curriculum as a graph search problem is not novel. Additionally, two classic search algorithms (GBFS & MCTS) studied to make the contribution of the work rather limited.
2. The evaluation baselines are sparse and do not include comparisons with more advanced methods in relevant areas, such as Curriculum Learning and Transfer Learning. Moreover, the domains of the evaluation tasks are limited, with most experiments focused on NLP tasks and the benchmarks used not being particularly advanced.
3. Important details about the method are missing, making it difficult to fully understand its implementation.

**Questions:**

1. What distinguishes TaCL from LoRA, particularly in the context of task adaptation for popular large language models (LLMs)?
2. How are the intermediate tasks designed—are they generated or pre-designed? Additionally, what does Q(v’) represent in Equation 6?
3. How does TaCL perform in CV or robotics tasks? How does it compare to other advanced methods?

---

> ### Author Response · Authors · 2024-11-26
>
> Thank you for your comments! Please find our responses below.
>
> > **Q1**: What distinguishes TaCL from LoRA?
>
> **A1**: While PEFT methods such as LoRA are popular, they aim to reducing the training parameters of large models. Instead, TaCL focuses on developing a transfer learning strategy that determines a sequence of training tasks, i.e., a curriculum, to improve the target task performance. Our experiments focus on full model training but it can also extend to LoRA training. Since most baselines from previous works are for full model training, we chose to follow the same setting for fair comparisons.
>
> > **Q2**: How are the intermediate tasks designed—are they generated or pre-designed? What does Q(v’) represent in Equation 6?
>
> **A2**: The intermediate tasks are determined by a tree search algorithm on a graph of tasks. In our experiments, they are representative NLP tasks selected from well-established benchmarks. We did not generate or design these tasks ourselves. In Equation 6, $Q(v')$ represents the Q-value of a child node $v'$ (a candidate for the next intermediate task) in MCTS. This value reflects the estimated utility or reward of selecting the next training task during the search process, guiding the adaptive exploration of tranfer task sequences.
>
> > **Q3**: How does TaCL perform in CV or robotics tasks?
>
> **A3**: The experiments in this paper primarily focuses on the realm of NLP tasks, which is a large area of interest with numerous tasks to explore. But the proposed method is not domain-specific and can be applied to other domains such as CV and robotics, which will be explored in our following works.

---

### Official Review · Reviewer_21dm · 2024-11-03

**Soundness:** 3
**Presentation:** 3
**Contribution:** 2
**Rating:** 3
**Confidence:** 5

**Summary:**

This paper introduces Task-Adaptation Curriculum Learning (TACL), a method to improve model adaptation to resource constrained target tasks by identifying and adapting model to intermediate tasks in a curriculum learning setting. The motivation is to mitigate the over-fitting issues that could rise when the amount  of target data is limited and is also characterized  by a large distribution	shift from the pre-training datasets. The authors propose to use existing publicly available datasets to define appropriate intermediate tasks and adapt the model thus battling the limited data issue.
To this end, authors forms this problem as a graph search problem, where each task is represented as a node. Their approach identifies an optimal sequence of tasks by evaluating task transferability using two search algorithms: Greedy Best-First Search (GBFS) and Monte Carlo Tree Search (MCTS). GBFS makes local, stepwise choices for each task in the sequence, while MCTS explores the sequence space more broadly, balancing exploration and exploitation via simulations by posing it as a multi-armed bandit problem. To estimate task-transferability, they first adapt the model on the intermediate task and then evaluate on the target task and measure heuristics such as validation loss or accuracy.  Furthermore, to reduce high computational costs, the authors propose to  limit the training steps on intermediate tasks to make a quick approximation of task-transferability.
They have conducted experiments on a 20-task and 6-task graphs with NLP benchmarks. Since their approach requires a pre-determined  graph, for the 20-task case, they compute it using previous studies and also prune the complete graph to reduce the search space.  Their experiments demonstrate that TACL significantly outperforms  naive fine-tuning and even a random order of tasks. MCTS seems to perform better most of the time.  Overall, TACL presents an effective approach to bridging gaps between pre-trained and target tasks, enhancing model generalizability across diverse task types.

**Strengths:**

The paper is well-written and easy to follow. The motivation is sound, and this is an important direction as the community increasingly moves toward fine-tuning from pre-trained models rather than training from scratch. Framing the problem of identifying intermediate tasks as a graph search selection is both interesting and a well-founded choice.

**Weaknesses:**

I have a few concerns and questions regarding the approach. First, there is a requirement for predetermined graphs, at least for GBFS. Could the authors clarify how they obtained the graph for the 6-task setting? They have explained their approach for the 20-task graph, but it’s not immediately clear how they obtained the 6-task graph. Was it generated in an almost brute-force manner, where the neighbors of a node include all tasks in the graph? Clarification on this point would be appreciated.

A major concern is that, in many domains, a predetermined task graph might not be readily available. It is also unclear how to address this issue in such settings. Additionally, I suggest that the authors consider augmentation-based baselines that address data scarcity issues or use generative models like LLMs (e.g., from [https://arxiv.org/pdf/2403.02990](https://arxiv.org/pdf/2403.02990)).

Furthermore, the idea of using similar tasks and discovering task relationships is well-studied in computer vision. For example, the CVPR 2018 best paper award-winning work on Taskonomy ([http://taskonomy.stanford.edu/](http://taskonomy.stanford.edu/)) addresses a similar problem and reveals a task graph. Please consider citing this work and discussing the connections.

The proposed solution also resembles meta-learning but lacks a meta-test update. Specifically, similar to meta-training, TACL adapts on an intermediate task, then evaluates this adaptation on the target task, akin to meta-testing. Meta-learning would use both gradients for updates, while TACL uses a simpler approach. Another relevant work, [https://arxiv.org/pdf/1911.10600](https://arxiv.org/pdf/1911.10600), addresses a similar issue and uses meta-learning to reveal the graph of task relationships, scaling to as many as 400 tasks. I recommend discussing these approaches.

I also suggest the authors comment on, or experiment with, anti-curriculum learning (i.e., training with harder tasks first). Studies such as [https://arxiv.org/abs/1707.09533](https://arxiv.org/abs/1707.09533) and [https://arxiv.org/abs/1811.00739](https://arxiv.org/abs/1811.00739), show that anti-curriculum learning can sometimes outperform standard curriculum learning.

Additionally, I am concerned that reducing the number of training steps may not be ideal for estimating the transferability score. Deep networks often exhibit grokking behavior ([https://arxiv.org/abs/2201.02177](https://arxiv.org/abs/2201.02177)) and double descent. It would be helpful to see a comparison or discussion on how these phenomena might impact the transferability scores.

A very minor point is that in a resource-constrained setting, the validation set is limited by definition, and I wonder if the heuristics are meaningful, given that they carve a portion from the training data.

Finally, there is a strong connection between TACL and continual learning. For instance, [https://arxiv.org/abs/2205.13323](https://arxiv.org/abs/2205.13323) examines the impact of task ordering in continual learning and proposes curriculum learning. Expanding the related work to include connections to continual learning would strengthen the paper.

One last critical piece missing is a baseline that updates only part of the network rather than the entire model, such as using LoRA. This approach might reduce overfitting by limiting the number of updated parameters. I suggest exploring this experiment, even for the 6-task setting, as parameter-efficient tuning is becoming as common as fine-tuning entire pre-trained models.

**Questions:**

Please see weaknesses section. I have listed the questions there as well.

---

> ### Author Response · Authors · 2024-11-26
>
> Thank you for your constructive feedback! Here are the responses to your questions.
>
> > **Q1**: How did authors obtain the 6-task graph?
>
> **A1**: For the 6-task graph experiments, we selected 6 representative tasks from the GLUE benchmark and assumed no prior information on the transferability between these tasks, i.e. we simply use a complete graph structure, meaning that every task is conntected to all other tasks. Our experiments demonstrate the effectiveness of our method when no prior information is available.
>
> > **Q2**: How to address the settings where predetermined task graph are not available?
>
> **A2**: When such information is not available, it might be computationally costly to conduct search on a fully-connected graph. In practice, however, we can efficiently estimate transferability using existing training-free or light-weight finetuning based methods to obtain such a prior graph structure.
>
> > **Q3**: Can authors comment on the Taskonomy, meta learning, anti-curriculum learning, and continual learning?
>
> **A3**:
> - **Taskonomy** aims to estimate the relationships between tasks, which are utilized in our method. But we target at an entirely different problem. We focuses on finding a curriculum of tranfer learning tasks for a target task by leveraging task transferbility information, rather than proposing an entirely new method to estimate task relationships/transferability.
>
> - **Anti-curriculum** uses a pre-defined hard-to-easy curriculum that prioritizes to learn difficult data samples at first. In contrast, our method is a task-level adaptive curriculum: (1) the curriculum is a curated sequence of training tasks instead of instances; (2) we determine the task per stage in an adaptive and dynamic manner by efficient tree-search based on the model in each stage.
>
> - **Continual learning** is different from curriculum learning studied in this paper on both the problem setting and the goal: (1) continual learning cannot control the order of learning tasks while curriculum learning focus on finding the best order; (2) continual learning aims to maintain the knowledge of all learned tasks but our goal is to achieve better transfer learning performance on the final target task.
>
> - **Meta-learning** studies a different problem as our transfer learning. They train a task-agnostic meta-model on training tasks and apply it to test tasks, where both the training and test tasks are drawn from the same distribution. In contrast, we do not train any meta-model and we apply the resulting model to the target task that is different from any training task in the curriculum. Transferability between tasks is interesting to both meta-learning and transfer learning communities.
>
> > **Q4**: Can authors provide comparison with LoRA?
>
> A4: While PEFT methods such as LoRA are popular, they aim to reducing the training parameters of large models. Instead, we focus on developing a transfer learning strategy that determines a sequence of training tasks, i.e., a curriculum, to improve the target task performance. Our experiments focus on full model training but it can also extend to LoRA training. Since most baselines from previous works are for full model training, we chose to follow the same setting for fair comparisons.

---

### Note · Authors · 2024-12-18

I have read and agree with the venue's withdrawal policy on behalf of myself and my co-authors.